# Genome-Wide Methylation Analysis in Two Wild-Type Non-Small Cell Lung Cancer Subgroups with Negative and High PD-L1 Expression

**DOI:** 10.3390/cancers16101841

**Published:** 2024-05-11

**Authors:** Georg Hutarew, Beate Alinger-Scharinger, Karl Sotlar, Theo F. J. Kraus

**Affiliations:** Institute of Pathology, University Hospital Salzburg, Paracelsus Medical University, Müllner Hauptstr. 48, A-5020 Salzburg, Austria; b.alinger@salk.at (B.A.-S.); k.sotlar@salk.at (K.S.); t.kraus@salk.at (T.F.J.K.)

**Keywords:** lung cancer, epigenetic profiling, methylome, methylation analysis, programmed cell death ligand 1, precision medicine

## Abstract

**Simple Summary:**

PD-L1 is a marker that helps determine a tumor’s immune status and is used to select patients for immune therapy. Methylation modifies gene expression by adding a methyl group to DNA without affecting its sequence. In our study, we assessed and correlated PD-L1 and methylation status in pulmonary adenocarcinomas with high and negative PD-L1 expression. We investigated the pathobiological functions of the highest-ranking genes and promoters in both groups by searching genomic databases for their role in carcinomas. We observed distinct methylation patterns between PD-L1 high- and low-expressing tumors, indicating differences in their biological characteristics and tumor development.

**Abstract:**

We conducted a pilot study to analyze the differential methylation status of 20 primary acinar adenocarcinomas of the lungs. These adenocarcinomas had to be wild type in mutation analysis and had either high (TPS > 50%; *n* = 10) or negative (TPS < 1%; *n* = 10) PD-L1 status to be integrated into our study. To examine the methylation of 866,895 specific sites, we utilized the Illumina Infinium EPIC bead chip array. Both hypermethylation and hypomethylation play significant roles in tumor development, progression, and metastasis. They also impact the formation of the tumor microenvironment, which plays a decisive role in tumor differentiation, epigenetics, dissemination, and immune evasion. The gained methylation patterns were correlated with PD-L1 expression. Our analysis has identified distinct methylation patterns in lung adenocarcinomas with high and negative PD-L1 expression. After analyzing the correlation between the methylation results of genes and promoters with their pathobiology, we found that tumors with high expression of PD-L1 tend to exhibit oncogenic effects through hypermethylation. On the other hand, tumors with negative PD-L1 expression show loss of their suppressor functions through hypomethylation. The suppressor functions of hypermethylated genes and promoters are ineffective compared to simultaneously activated dominant oncogenic mechanisms. The tumor microenvironment supports tumor growth in both groups.

## 1. Introduction

Lung cancer is a widespread form of cancer. The chances of surviving NSCLC vary from 19 to 63% depending on the stage and available treatment options [1,2]. Modern oncology offers many drugs designed to treat specific targets; therefore, a molecular panel that includes immunohistochemistry and next-generation sequencing (NGS) with RNA and DNA analyses is mandatory for each pulmonary carcinoma [3]. In routine lung cancer diagnosis, we have to detect defined mutations in genes such as *EGFR*, *BRAF*, *KRAS*, *ERBB2*, and *STK11* and defined rearrangements like *ALK*, *RET*, *ROS1*, *NTRK*, and *cMET* [4]. In a study with real-world data analysis, the median overall survival was 351 days for all patients and 571 days for those with targetable activating mutations and/or activating translocations [5]. Many studies outline the advances and improved therapy outcomes of target therapy in general or for specific targets and drugs, such as EGFR TKIs and ALK inhibitors, in patients with pulmonary adenocarcinomas [6,7,8,9,10,11].

A significant advancement in oncology was the development of immunotherapy with checkpoint inhibitors (ICIs). In lung cancer, PD-1 and PD-L1 inhibitors have shown promising outcomes for selected patients based on their PD-L1 status, with partial or even complete tumor regression in the case of pulmonary adenocarcinomas [12,13,14]. ICIs like pembrolizumab, nivolumab, atezolizumab, and durvalumab were used initially in second-line therapy and later in first-line therapy [15,16]. Meanwhile, they are available for consolidation therapy following chemoradiation in unresectable locally advanced diseases and also in neoadjuvant settings after surgical resection and chemotherapy [17,18,19]. Few patients become long-term survivors with a chronification of lung cancer [20]. Patients are selected for therapy with the expression of PD-L1 antibodies. Programmed death ligand 1 (PD-L1) is a transmembrane protein expressed on the surface of T cells, NK cells, tumor-associated macrophages (TAMs), myeloid dendritic cells, and epithelial cells. Interaction between PD-L1 and its receptor PD-1 modifies the immune response in the tumor microenvironment, particularly by decreasing the effectiveness of physiological anti-tumor mechanisms [21] and by preventing the activation of cytotoxic T-cells [22,23]. Cancer cells upregulate PD-L1 on their surface to evade immune-regulated destruction by cytotoxic T-cells [24]. Blocking the interaction of PD-L1 and PD-1 with checkpoint inhibitors can enable cytotoxic T-cells to destroy tumor cells again.

PD-L1 expression and interactions play a crucial role in immune modulation, particularly in cancer contexts. The same is true for DNA methylation, as well as for chemotherapy and target therapy [25,26]. Changes in the immune status result in changes in the methylation status in large tumor regions [27]. Methylation refers to the addition of a methyl group to DNA without altering its sequence. It mainly influences gene expression and is the reason for epigenetic regulation, especially in cancer, which also impacts the tumor microenvironment.

We conducted a genome-wide comprehensive epigenomic methylation analysis to identify methylation patterns in lung cancer with different PD-L1 statuses (i.e., PD-L1 high, TPS > 50%, and PD-L1 negative, TPS < 1%). To avoid the influence of aberrant DNA methylation from mutations and rearrangements [28] in our cohort of lung cancer patients, we only included cases that were analyzed with next-generation sequencing as wild type and without rearrangements.

We aimed to find the pathobiological mechanisms and functions in PD-L1 high- and negative-expressing pulmonary adenocarcinomas based on genes and promoters with significantly different levels of methylation in both groups. We chose specific cutoffs and selected the genes and promoters with the highest and lowest rankings from both groups. We consulted multiple databases, including GeneCards, the National Institutes of Health (NIH), the Alliance of Genome Resources, and The Cancer Genome Atlas (TCGA) to gather data on their pathobiological functions.

The retrieved pathobiological data and functions enabled us to develop a hypothesis on how methylation may affect the pathobiological mechanisms of pulmonary adenocarcinomas with negative and high PD-L1 expression.

## 2. Materials and Methods

### 2.1. Tissue Collection and Immunohistochemical Analysis

Our study used formalin-fixed and paraffin-embedded (FFPE) archived samples from the University Institute of Pathology, University Hospital Salzburg, and Paracelsus Medical University, Salzburg, Austria. Cases were randomly selected from our routine material, but they had to meet specific criteria. Each tumor was required to be a primary pulmonary adenocarcinoma, exhibiting either high PD-L1 expression (TPS > 50%) or negative PD-L1 expression (TPS < 1%), with no mutations or rearrangements in the target genes or gene regions according to our next-generation sequencing data (please see below for details). Figure 1 illustrates the distribution of the PD-L1 values.

The level of PD-L1 expression in lung cancer is determined by calculating the tumor proportion score (TPS) based on guidelines set by the US Food and Drug Administration (FDA) and the National Comprehensive Cancer Network (NCCN) [29,30]. This score is used as a biomarker to predict the probability of a positive response to immune therapy in lung cancer patients [30,31]. TPS is the percentage of stained viable tumor cells with complete or partial membranous staining at any intensity falling into three groups: negative TPS < 1%, low TPS 1–49%, and high TPS ≥ 50%. The higher the TPS, the better the individual’s response to checkpoint inhibitor therapy is assumed [32]. In recent years, pathologists have encountered a number of inconsistencies in testing PD-L1, which has led to its reputation as a fragile biomarker [33]. Tumors exhibit a heterogeneous staining pattern, with varying levels of PD-L1 expression across different regions, making it challenging to assess the overall PD-L1 status. The quality of results can vary significantly due to differences in the PD-L1 antibodies used, staining protocols and systems, and inter- and intraobserver variability. This variation can significantly affect the scoring and treatment decisions of non-small cell lung cancers (NSCLCs). However, it can be minimized by relying on expertise, routine testing, and using a validated PD-L1 antibody. To determine the PD-L1 status of NSCLCs, we used the DAKO 22C3pharm DX kit from Agilent, USA, which is a companion diagnostic tool. The immunostainings were performed on a DAKO-Omnis System (Agilent, Santa Clara, CA, USA). Finally, we selected 10 cases with high PD-L1 expression (TPS > 50%, mean 82%; *n* = 10) and 10 patients with negative PD-L1 expression (TPS < 1%, mean 0.5%; *n* = 10). Figure 2A,B and Figure 3A,B show HE stains and PD-L1 stains of a negative- and high-expressing adenocarcinoma, respectively.

### 2.2. Molecular and Genetic Analysis

The methods used in this study for DNA and RNA extraction and analysis are based on our routine specimen workup, from which our probes are derived. To ensure accurate mutation analysis, tumor specimens with a minimum of 60% tumor content were used. In some cases, we had to increase the amount of tumor tissue for analysis by microdissecting the specific area of interest on the tissue slides using a simple and effective method. We marked the areas with a higher concentration of tumor cells on the HE slide. Then, we scratched off the corresponding material from the following native cutting planes to use in the subsequent analysis. DNA was extracted from FFPE tissue samples using the Maxwell RSC (Promega, Fitchburg, WI, USA) automated purification system, which includes an overnight Proteinase K Digestion step prior to DNA purification. The DNA concentration was measured with a Quantus™ Fluorometer that allows an automatic concentration calculation. It is particularly effective for analyzing formalin-fixed paraffin-embedded (FFPE) tissue samples with low levels of nucleic acid. RNA was extracted from FFPE samples using the RNeasy FFPE Kit from Qiagen^®^ (Hilden, Germany). This kit is specifically designed to extract RNA molecules longer than 70 nucleotides and provides usable RNA fragments for various downstream applications. The kit uses special lysis and incubation conditions to reverse the formaldehyde modification of RNA. The lysis buffer efficiently releases RNA from FFPE tissue sections, minimizing further RNA degradation. The RNA concentration was measured with a Quantus™ Fluorometer. We used the Illumina AmpliSeq cDNA Synthesis Kit to convert RNA into cDNA. This kit is specifically designed for use with library preparation kits and panels for targeted sequencing.

For mutation analysis of lung adenocarcinomas, we used either the AmpliSeq for Illumina Cancer Hotspot Panel v2 or the AmpliSeq for Illumina Focus Panel (Illumina, San Diego, CA, USA).

The AmpliSeq for Illumina Cancer Hotspot Panel v2 is a targeted resequencing assay specially designed to investigate somatic mutations, such as single-nucleotide polymorphisms (SNPs), somatic variants, and insertions–deletions (indels), across the hotspot regions of 50 genes that are known to be associated with cancer, particularly in solid tumors. The AmpliSeq for Illumina Focus Panel is a targeted resequencing assay designed for biomarker analysis of 52 genes. It can analyze both DNA and RNA simultaneously and can detect copy number variants (CNVs), gene fusions, insertions–deletions (indels), single-nucleotide polymorphisms (SNPs), and somatic variants. For both assays, the input recommendation for DNA or RNA is 1–100 ng (10 ng per pool). The workflow starts with a polymerase chain reaction (PCR)-based library preparation in which DNA or cDNA fragments are amplified, primer sequences are removed, and the fragments are ligated to adapters. After PCR amplification with sequence primers, the fragments can be sequenced by synthesis (SBS) next-generation sequencing (NGS) technology and data analysis with the onboard software tool. Next-generation sequencing was performed on an Illumina MiniSeq device following the manufacturer’s protocol.

To analyze aberrant DNA methylation patterns (hyper- and hypomethylation) in PD-L1 high- and negative-expressing pulmonary adenocarcinomas, we employed the Infinium Methylation EPIC Bead Chip with Infinium chemistry and the Illumina protocol. The kit detects cytosine methylation at CpG islands, providing broad coverage of genes and enhancers for epigenome-wide association studies (EWAS) [34]. It enables precise and accurate measurement of methylation at single-nucleotide resolution for CpGs. Infinium chemistry detects cytosine methylation through highly multiplexed genotyping of bisulfite-converted genomic DNA. Bisulfite conversion is essential for studying DNA methylation patterns where unmethylated cytosines are converted to uracils while methylated cytosines are preserved.

Illumina Genome Studio Methylation Module Software (Version 2011.1, Illumina, San Diego, CA, USA) has integrated controls that assess data quality based on Infinium workflow steps. These include sample-specific controls, such as the efficiency of bisulfite conversion, and negative controls, which we used for primary controls. All other statistical and quality controls are integrated into the RnBeads package (please see below).

### 2.3. Computational Data Analysis

There were various reasons why we utilized RnBeads (Package 2.18.0) on R statistical software (Version 4.3.0, R Foundation for Statistical Computing, Vienna, Austria) [35] to analyze our methylation data. RnBeads offers a more comprehensive analysis workflow than other existing tools. It also provides state-of-the-art normalization techniques that enable robust sample comparisons. Additionally, experimental quality control is integrated into the package and can be conducted to identify sample outliers and mix-ups. RnBeads is designed to characterize differential methylation between groups of samples. In our study, we compared PD-L1 high-expressing samples to negative-expressing samples. It analyzes individual CpGs and allows sample filtering on predefined or custom genomic regions. The generated methylation data reports include method descriptions, plots, and data tables.

Our mapping was based on the human genome reference builds (GRCh38.p14 patch release of the hg38 assembly). DNA methylation beta values were used, indicating the ratio of the intensity of the methylated bead type in variables between 0 and 1, depending on the combined locus intensity (ranging from 0% to 100% methylation). To improve the reliability of our results, we removed probes enriched with single-nucleotide polymorphisms (SNPs) and used Greedycut filtering to remove measurements with the highest impurity. During an iterative algorithm, unreliable measurements with *p* > 0.05 were eliminated based on detection *p*-values. A β value is considered unreliable if its corresponding *p*-value is not below 0.05. We also removed context-specific probes and those on sex chromosomes. We utilized the Dasen Method to remove technical artifacts such as background fluorescence, dye bias, probe design bias, and batch effects from our DNA methylation data. This normalization technique adjusts raw intensities to enhance the comparability of DNA methylation data between different samples [36]. Figure 4A shows the graphical distribution of removed and retained methylation β values. Figure 4B demonstrates Dasen normalization’s effect on CpG methylation values. Probes were marked with four genomic areas, including tiling regions (which were 5000 nucleotides long), genes (version Ensemble genes 75), promoters (version Ensemble genes 75), and CpG islands (CpG island track of the UCSC Genome Browser) [37]. We used a hierarchical linear model (Limma package) to analyze gene expression data and assess differential expression with associated *p*-values and false discovery rates with an empirical Bayes approach, which is a procedure for statistical inference in which the prior probability distribution is estimated from the data [38]. To identify sites exhibiting differential variability between two sample groups, we used the diffVar method integrated into the missMethyl package and nominal *p*-values (nominal significances) as performed in other epigenome-wide association studies (EWAS) [39,40].

In summary, RnBeads is a package that adheres to the three customary procedures for computational analysis of DNA methylation data, which include (I) data processing and quality control, (II) data visualization and statistical analysis, and (III) validation and interpretation. The study setup involves conducting differential methylation analysis using a strict protocol, followed by researching the pathobiological functions of the highest- and lowest-ranking results within predefined cutoffs. This process, therefore, should be easily reproducible.

## 3. Results

In the Section 3, we present the technical data from our methylation analysis study, whereas the pathobiological correlation and functions were integrated into the Section 4 as we hope for better understanding. Our analysis focused on identifying the methylation status between PD-L1 high adenocarcinomas (TPS > 50%) and PD-L1 negative cases (TPS < 1%) without any known relevant gene mutation or rearrangement. We analyzed 20 samples with 866,895 methylation sites and performed statistical analyses using RnBeads, as detailed in Section 2.3 “Computational Analysis”. We had to exclude 17,371 sites due to overlapping with SNPs and 7532 sites after applying the Greedycut algorithm. Additionally, we removed 18,597 probes located on sex chromosomes and 2915 context-specific probes. We kept all samples but, in the end, removed 46,415 probes, thus retaining 820,480 probes for final analyses, as depicted in Figure 4A. Our analysis identified 252,729 annotations in the tiling regions (length 5000), 34,988 annotations in genes, 44,852 annotations in promoters, and 26,540 annotations in CpG islands.

Based on the average methylation levels, we divided the results for genes and promoters into two categories: highest average (mean.mean.high) and lowest average (mean.mean.low). Additionally, we identified the highest differential methylation (mean.mean.diff) for genes and promoters between the two groups with *p*-values less than 0.05 (*p* < 0.05). Figure 5A,B illustrate the graphical distribution of the results. The scatterplot shows the mean methylation difference between the PD-L1 high and low groups, and the volcano plot compares methylation between PD-L1 high and negative groups using pairwise comparisons. To choose genes and promoters from these categories, we used a normal (Gaussian) distribution. We selected those with values more than two standard deviations above (corresponding to >95%) or below (corresponding to <5%) the mean value at 50% [41]. We conducted literature research on the selected genes and promoters using databases like GeneCards (https://www.genecards.org/, accessed on 1 March 2024), the National Institutes of Health (NIH) (https://www.nih.gov/, accessed on 1 March 2024), the Alliance of Genome Resources (https://www.alliancegenome.org/, accessed on 1 March 2024), and The Cancer Genome Atlas (TCGA) (https://www.cancer.gov/ccg/research/genome-sequencing/tcga, accessed on 1 March 2024). Our aim was to identify the ones that are associated with lung cancer pathobiology within this group. Table 1 provides the detailed outcomes of the results.

We have organized the highest-ranking results into four different groups:

Section 3.1 (Hypermethylated genes and promoters in PD-L1 high-expressing cases);

Section 3.2 (Hypomethylated genes and promoters in PD-L1 high-expressing cases);

Section 3.3 (Hypermethylated genes and promoters in PD-L1 low-expressing cases);

Section 3.4 (Hypomethylated genes and promoters in PD-L1 low-expressing cases).

### 3.1. Hypermethylated Genes and Promoters in PD-L1 High-Expressing Cases

Average hypermethylation in this group ranged from 80% to 90%, with *p*-values from 0.01 to 0.04 in the following locations:

*SNORD114-14* (C/D Box 114-14, Small Nucleolar RNAs, snoRNAs), *DCAF4L2* (DDB1 Associated Factor 4 Like 2), *CELF2-AS1* (CELF2 Antisense RNA 1), *LINCMD1* (Long Intergenic Non-Protein Coding RNA, Muscle Differentiation, MIR133BHG).

The most significant differences in methylation (delta value) between PD-L1 high- and low-expression groups were found in the following:

*S100A7L2* (S100 Calcium Binding Protein A7 Like 2) and *SOD1P3* (Superoxide Dismutase 1 Pseudogene 3).

### 3.2. Hypomethylated Genes and Promoters in PD-L1 High-Expressing Cases

Average hypomethylation was found at rates ranging from 14% to 21%, with *p*-values between 0.007 and 0.03 in the following locations:

*CAPS2* (Cyclase-Associated Protein 2, Calcyphosine 2), *GLIPR1L2* (GLI Pathogenesis Related 1 Like 2), and *IFITM3* (Interferon Induced Transmembrane Protein 3).

### 3.3. Hypermethylated Genes and Promoters in PD-L1 Negative-Expressing Cases

Average hypermethylation ranged from 73% to 90%, with *p*-values between 0.001 to 0.04 in the following locations:

*SNORD114-14* (C/D Box 114-14, Small Nucleolar RNAs, snoRNAs) and *LINC00528* (Long Intergenic Non-Protein Coding RNA528).

### 3.4. Hypomethylated Genes and Promoters in PD-L1 Negative-Expressing Cases

Average methylation in PD-L1 negative-expressing cases between 21% to 30% and with *p*-values between 0.005 to 0.02 was found in the following:

*MIR124-3* (MicroRNA124-3), *TRIM71* (Tripartite Motif Containing 71, LIN41), *CAPS2* (Calcyphosine 2), *UBE2QL1* (Ubiquitin Conjugating Enzyme E2 Q Family Like 1), and *GLIPR1L2* (GLIPR1 Like 2).

The highest methylation delta values between PD-L1 low/high were found in the following:

*NUMB* gene (NUMB Endocytic Adaptor Protein) and *LINC00528* (Long Intergenic Non-Protein Coding RNA 528).

Additionally, we found a group with the simultaneous methylation status of genes and promoters in PD-L1 high- and negative-expressing carcinomas, suggesting independence from the PD-L1 status. *SNORD114-14* showed the highest levels of methylation in both PD-L1 high- and PD-L1 negative-expressing lung cancers, with a methylation of 75% and 90%, respectively, and *p*-values of 0.04. Moreover, *CAPS2* and *GLIPR1L2* were hypomethylated in both PD-L1 high and negative expression groups with a methylation of 14%, 17% (genes), and 26% (promoter) and *p*-values of 0.005 for *CAPS2* and 19% and 30% and *p*-values of 0.008 for *GLIPR1L2*.

## 4. Discussion

The expression of genes can differ due to various epigenetic mechanisms cells use to regulate DNA functions. One of these mechanisms is DNA methylation, which alters gene expression without changing its sequence. This modification can affect gene expression during cell differentiation. Methylation of a promoter region can regulate nearby gene expression, and excessive methylation can cause the silencing mainly of DNA repair genes [42]. This is believed to be an early step toward cancer progression, as hypermethylation of DNA upstream blocks access to transcription factors and enzymes, ultimately inhibiting downstream gene activity [43]. Conversely, many tumors exhibit hypomethylated carcinogenic genes when compared to normal tissue [44,45]. Activating typically silenced genes can contribute to developing malignant neoplasias as individuals age. Both hypermethylation and hypomethylation are leading causes of oncogenesis, the former being more frequent and occurring at the CpG islands in the promoter region of the genes. In contrast, the latter occurs globally in various genomic sequences [46].

Several genes exhibit significant methylation status, directly influencing angiogenesis, active oxygen, calcium, and vessels closely related to the tumor microenvironment [47]. Additionally, the methylation status of several genes and promoters is associated with modifying the status of immune cells such as macrophages, lymphocytes, and neutrophils [48]. These genes and promoters help to create an optimized microenvironment for tumor growth and development [42,49].

Some studies have examined the impact of PD-L1 methylation on PD-L1/PD-1 interactions, as PD-L1 expression is known to influence the immune status. Epigenetic regulations, such as methylation and histone acetylation, determine the expression levels of PD-L1. In the case of pancreatic carcinoma, these regulations cause an increase in PD-L1 expression, while in hepatocellular carcinoma, head and neck squamous cell carcinomas, and other types of cancer, the expression of PD-L1 is decreased [22]. PD-L1 DNA methylation has a functional relationship with mRNA expression in NSCLC. However, the correlation between methylation and PD-L1 expression in tumor biopsies was inconclusive [50].

PD-L1 K162 methylation inhibits PD-L1/PD-1 binding, preventing tumor immune escape even with high PD-L1 expression. PD-L1 hypermethylation was shown to be a key mechanism for anti-PD-L1 therapy resistance [51]. In our study, the methylation status of PD-L1, PD-L1 K162, and PD-1 did not show significance. Although these questions are intriguing, they were not included in this study’s scope.

Our research focused on characterizing the pathobiological functions of PD-L1 high- and negative-expressing pulmonary adenocarcinomas based on the results of our methylation analysis.

### 4.1. Pathobiological Mechanisms of Hypermethylated Genes and Promoters in the PD-L1 High-Expressing Tumors

*SNORD114-14* (C/D Box 114-14, Small Nucleolar RNAs, snoRNAs) (average methylation 90%, *p* = 0.04) and other small nucleolar RNAs (snoRNAs) are crucial in developing and spreading lung cancer. They control the balance between cell growth and death and promote the adaptability of cancer cells. SnoRNAs display both oncogenic and tumor-suppressive activities that are vital in the formation and progression of lung cancer. Dysregulation of snoRNAs is a contributing factor to lung cancer tumorigenesis and progression [52].

*DCAF4L2* (DDB1 Associated Factor 4 Like 2) (average methylation 82%, *p* = 0.02) is a protein-coding gene. Studies have revealed that the levels of DCAF4L2 are higher in patients with lung cancer and colorectal cancer, which can lead to more advanced stages of the disease and the spread of cancer cells to other parts of the body [53].

*CELF2-AS1* (CELF2 Antisense RNA 1) (average methylation 80%, *p* = 0.02) suppresses non-small cell lung carcinoma growth by inhibiting the PREX2-PTEN interaction, which regulates cell proliferation. Additionally, CELF2 protein expression is downregulated in tumor tissues as lung cancer and is associated with poor prognosis [54].

*LINCMD1* (Long Intergenic Non-Protein Coding RNA, Muscle Differentiation, MIR133BHG) (average methylation 80%, *p* = 0.01) inhibits cell proliferation, migration, and the invasion of lung adenocarcinoma [55,56].

Despite being classified as a pseudogene, *S100A7L2* (S100 Calcium Binding Protein A7 Like 2) (average methylation 74%, delta17%, *p* = 0.002) has been found to play a significant role in inducing transdifferentiation from lung adenocarcinoma to squamous carcinoma. Moreover, it has been observed that the expression of this gene is regulated differently by the Hippo-YAP pathway in lung cancer cells, which contributes to tumor cell growth [57]. *S100A7L2* is also involved in cell migration and invasion, creating a proinflammatory and proangiogenic environment that promotes tumor progression and metastasis [58].

The gene *SOD1P3* (Superoxide Dismutase 1 Pseudogene 3) (average methylation 68%, delta 14%, *p* = 0.02) substantially influences cancer development. *SOD3* and *SOD1P3* regulate active oxygen in the microenvironment and are downregulated in lung cancer [59]. Furthermore, *SOD1P3* regulates the processes of angiogenesis, metastasis, and invasion in lung cancer by controlling the levels of Interleukin6 and VEGF [60].

The summarized results show that methylation-driven genes and promoters in PD-L1 high-expressing tumors have strong oncogenic effects (*SNORD114-14*, *DCAF4L2*, *S100A7L2*, and *SOD1P3*). Additionally, hypermethylation in PD-L1 high-expressing tumors promotes the creation of a proinflammatory and proangiogenic milieu in the tumor microenvironment, associated with an increased number of immune cells such as neutrophils, macrophages, and lymphocytes, and regulates active oxygen and calcium [61]. On the other hand, hypermethylated genes and promoters in the PD-L1 high-expression group have suppressor effects (*CELF2-AS1*, *LINCMD1*, and *MIR133BHG*), which appear ineffective in reducing tumor progression compared to this group’s simultaneously activated dominant oncogenic mechanisms.

### 4.2. Pathobiological Mechanisms of Hypomethylated Genes and Promoters in the PD-L1 High-Expression Group

*CAPS2* (Calcyphosine 2) (average methylation 14%, *p* = 0.005) is overexpressed in lung cancer, where it promotes cell proliferation, migration, invasion, and metastasis by activating the ERK/MAPK and AKT signaling pathways [62]. The expression of CAPS2 is higher in lung cancer tissues when compared to normal lung tissues, and it is linked to the stage of the tumor, lymph node metastasis, and poor survival. As such, CAPS2 can be used as a biomarker to monitor lung cancer progression following therapy [63].

*GLIPR1L2* (average methylation 19%, *p* = 0.008) plays a significant role in cancer and immune defense. GLI pathogenesis-related 1 functions as a tumor suppressor in lung cancer, and during lung tumorigenesis, the expression of GLIPR1L2 is downregulated [64].

*IFITM3* (Interferon-Induced Transmembrane Protein 3) (average methylation 21%, *p* = 0.03) regulates the growth and invasion of human lung adenocarcinoma and is believed to be a crucial factor in promoting carcinogenesis. According to recent research, IFITM3 expression levels are directly associated with tumor differentiation, lymph node, distant metastasis, and tumor node metastasis stages. Knockdown of *IFITM3* is effective in suppressing lung cancer cell proliferation, invasion, and migration while inducing cell cycle arrest and apoptosis [65].

The results indicate that both these genes and promoters can exhibit oncogenic and tumor-suppressive effects. However, both functions seem to be reduced by hypomethylation, ultimately supporting the development of lung cancer.

### 4.3. Pathobiological Mechanisms of Hypermethylated Genes and Promoters in PD-L1 Negative-Expressing Cases

*SNORD114-14* (average methylation 90%, *p* = 0.04), its pathobiological mechanisms are already listed above.

*LINC00528* (Long Intergenic Non-Protein Coding RNA 528) (average methylation 73%, *p* = 0.001) was identified as a long non-coding RNA found in computational analyses to be associated with neutrophils, lymphocytes, and macrophages involved in the immune reaction of the tumor microenvironment and strongly correlated with immunotherapy prognosis in lung cancer. The exact mechanism of how *LINC00528* affects the immune system is not fully understood, but it is believed to regulate genes involved in the immune response [66,67].

### 4.4. Pathobiological Mechanisms of Hypomethylated Genes and Promoters in the PD-L1 Negative-Expressing Group

*MIR124-3* (MicroRNA124-3) (average methylation 21%, *p* = 0.01) is a microRNA that affects mRNA stability and translation. It plays a role in both breast and lung cancer [68] and was shown to suppress lung cancer metastasis [69].

*TRIM71* (Tripartite Motif Containing 71, LIN41) (average methylation 15%, *p* = 0.02) is a gene that encodes an E3 ubiquitin-protein ligase, which plays a role in the G1-S phase transition of the cell cycle. TRIM7 negatively regulates the NF-kappa B signaling pathway in lung cancer by degrading p65 [70,71]. The expression of TRIM7 is diminished in tumor tissues compared to adjacent normal tissues, and its level is negatively correlated with the clinical stage of lung cancer. In vitro, TRIM7 substantially inhibits the proliferation and migration of tumor cells and promotes cell apoptosis, but its effects are less effective in vivo [72].

*CAPS2* (Calcyphosine 2) (average methylation 26%, *p* = 0.005) and *GLIPR1L2* (GLI pathogenesis-related 1) (average methylation 30%, *p* = 0.008), both were previously discussed in the group due to hypomethylation in PD-L1 high-expressing carcinomas.

The *UBE2QL1* gene (Ubiquitin Conjugating Enzyme E2 Q Family Like 1) (average methylation 30%, *p* = 0.007) encodes a ubiquitin-conjugating enzyme in lung cancer cells. It plays a crucial role in regulating the integrity of lysosomes and controlling the selective macroautophagy or autophagy of the entire organelle, also known as lysophagy [73]. Lysosomal membrane permeabilization or complete rupture of lysosomes can lead to stress conditions relevant to degenerative diseases, infections, and cancer [74].

Based on several studies, the *NUMB* gene (NUMB Endocytic Adaptor Protein) (average methylation 51%, delta 20%, *p* = 0.005) acts as a tumor suppressor in lung carcinoma [75]. In lung adenocarcinoma, high levels of NUMB can inhibit tumor growth, invasion, the Notch pathway, and the epithelial-mesenchymal transition. However, in lung squamous cell carcinoma, *NUMB* may promote cell proliferation. The loss or mutation of *NUMB* is associated with poor prognosis, tumor progression, and resistance to chemotherapy in NSCLC patients [76].

We concluded that in this group, the process of hypomethylation leads to tumor development by reducing the effectiveness of the mechanisms that suppress tumor growth [77]. This suggests that tumors with high levels of PD-L1 expression in the lungs and other primary tumor sites increase oncogenic mechanisms, making them more aggressive than tumors with negative PD-L1 expression, which lose tumor-suppressing mechanisms [61].

It is important to note that our study was carried out on a limited sample size of only 20 cases, and we see it as a pilot study. The specimens were processed using robust and reproducible methods, including standardized PD-L1 immunostaining, methylation analysis from Illumina, and statistical evaluation with RnBeads. The applied cutoffs of 5% and 95% were chosen to reduce the number of evaluated genes and promoters. However, altering these cutoffs would increase the number of results and might influence the resulting pathobiology. Therefore, results should be interpreted with caution, and further research with a larger sample size should be performed to confirm our findings.

More research is necessary to explore the impact of methylation in upcoming studies on lung cancer. Methylation alterations have significant effects on tumor biology. Dynamic changes in methylation occurring during lung cancer progression and tissue-specific methylation patterns can serve as biomarkers for diagnosis or prediction. Besides chemotherapy, targeted therapy and immunotherapy, either alone or in combination, can lead to changes in genome-wide methylation patterns for monitoring disease progression or therapeutic outcomes.

## 5. Conclusions

PD-L1 high-expressing tumors show hypermethylated genes and promoters with strong oncogenic effects (*SNORD114-14*, *DCAF4L2*, *S100A7L2*, and *SOD1P3*) and hypermethylated genes and promoters with tumor suppressor effects (*CELF2-AS1* and *LINCMD1*). Hypomethylated genes and promoters in this group seem to lose their tumor-suppressing effects (*IFITM3* and *GLIPR1L2*) and perhaps show a reduction in oncogene activity (*CAPS2*).

PD-L1 negative-expressing tumors show a potent hypermethylated oncogene (*SNORD114-4*) and hypomethylated genes and promoters, which seem to lose their suppressor function (*MIR124-3*, *TRIM71*, *GLIPR1L2*, and *NUMB*) and additionally reduce the functionality of *CAPS2*, an oncogene. Moreover, *UBE2QL1* and *LINC0528* form a tumor microenvironment that supports cancer development, growth, and progression.

Lung carcinomas exhibiting high and negative PD-L1 expression demonstrate distinct methylation patterns, and their pathobiology indicates different paths through various mechanisms by which PD-L1 high-expressing and PD-L1 negative-expressing lung cancers develop. It appears that tumors with a high expression of PD-L1 are primarily driven by the development of oncogenic effects, and carcinomas with a negative expression of PD-L1 tend to develop tumors mainly by reducing suppressor mechanisms. If genes and promoters are hypermethylated, leading to the simultaneous upregulation of suppressor and oncogenic effects, suppressors seem less effective than dominant oncogenic mechanisms. We concluded that the activation of oncogenes correlates with more aggressive tumor behavior, as seen in the PD-L1 high group.

## Figures and Tables

**Figure 1 cancers-16-01841-f001:**
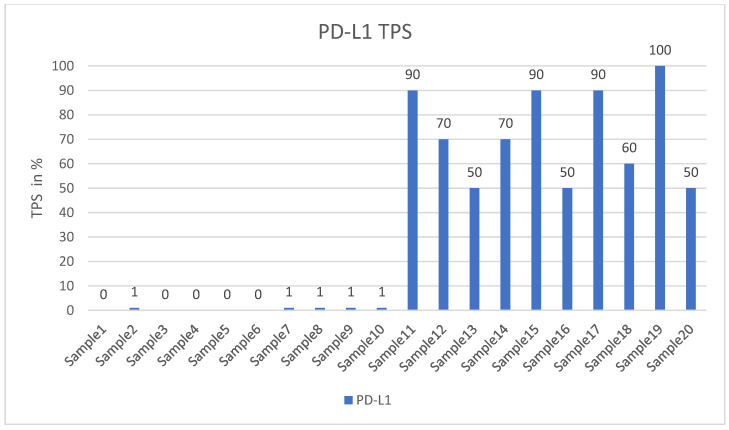
Shows the TPS values of all probes.

**Figure 2 cancers-16-01841-f002:**
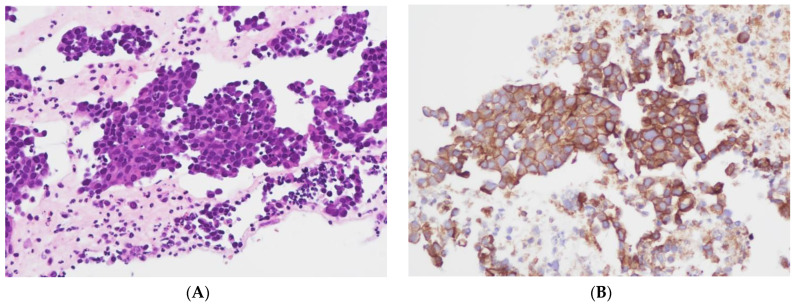
Cellblock of a pleural effusion with a pulmonary adenocarcinoma. (**A**): HE 200×. (**B**): PD-L1 (22C3) positive, TPS = 100% (Sample 19 in Figure 1) 200×.

**Figure 3 cancers-16-01841-f003:**
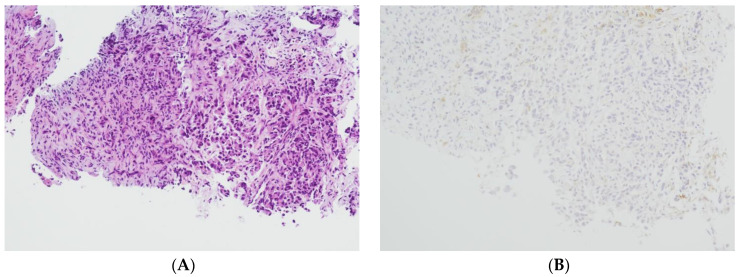
Lung biopsy fragments of a pulmonary adenocarcinoma. (**A**): HE 100×. (**B**): PD-L1 (22C3) negative, TPS = 0% (Sample 6 in Figure 1) 100×.

**Figure 4 cancers-16-01841-f004:**
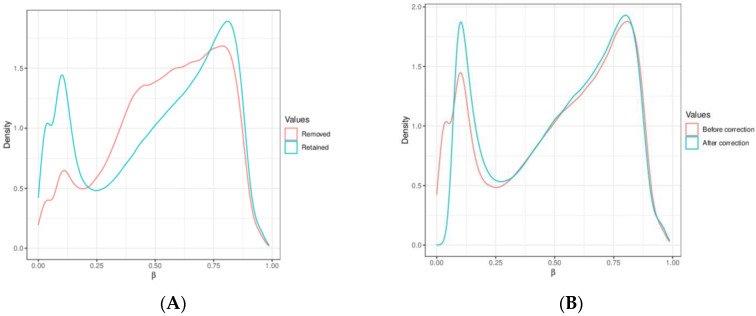
(**A**) Compares the methylation β values that were removed with those that were retained. (**B**) Demonstrates the effect of Dasen normalization on CpG methylation values, which reduces technical biases and enhances the reliability of DNA methylation data. The X-axis represents beta values (methylation level), while the Y-axis shows density (methyl group concentration) in specific genomic regions.

**Figure 5 cancers-16-01841-f005:**
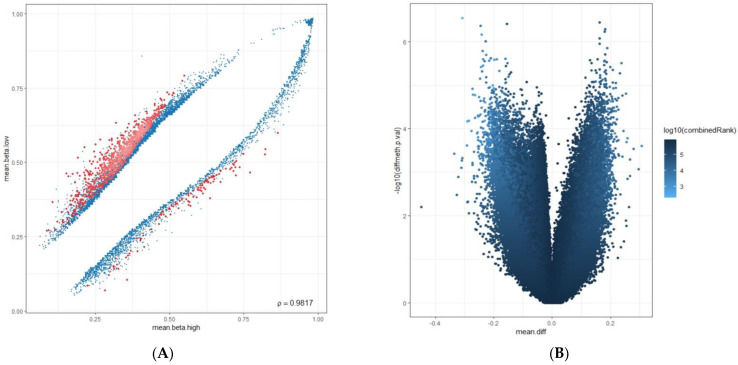
(**A**): The scatterplot shows the mean methylation difference between the PD-L1 high and low group in blue, with a combined rank among the top 1000 sites in red. The proportion of methylation (mean.beta.low and mean.beta.high) is used as a variable on the Y- and X-axes. (**B**): Volcano plots compare methylation between PD-L1 high and low groups using pairwise comparisons, with colors based on combined *p*-values to assess the overall significance. The X-axis shows the difference in average methylation (mean.diff), while the Y-axis shows the identification of differentially methylated CpG sites (−log10 (*p*-value)), (diff.meth).

**Table 1 cancers-16-01841-t001:** Methylation results.

PD-L1 Expression	Methylation Status	Gene	Methylation	*p* Value	Chr
PD-L1 high	Highest methylation in gene	*SNORD114-14* (C/D Box 114-14). Small Nucleolar RNAs (SnoRNAs)	0.9	0.04	14
	Highest methylation in promoter	*DCAF4L2* (DDB1 Associated Factor 4 Like 2)	0.82	0.02	8
		*CELF2-AS1* (CELF2 Antisense RNA 1)	0.8	0.02	10
		*LINCMD1* (Long Intergenic Non-Protein Coding RNA, Muscle Differentiation1, MIR133BHG)	0.8	0.01	6
	Lowest methylation in gene	*GLIPR1L2* (GLIPR1 Like 2)	0.21	0.007	12
	Lowest methylation in promoter	*CAPS2* (Calcyphosine 2).	0.14	0.005	12
		*GLIPR1L2* (GLI Pathogenesis Related 1 Like 2)	0.19	0.008	12
		*IFITM3* (Interferon Induced Transmembrane Protein 3)	0.21	0.03	11
PD-L1 negative	Highest methylation in gene	*SNORD114-14*, (C/D Box 114-14) Small Nucleolar RNAs (SnoRNAs)	0.75	0.04	14
		*LINC00528* (Long Intergenic Non-Protein Coding RNA 528)	0.73	0.001	224
	Lowest methylation in gene	*MIR124-3* (MicroRNA124-3)	0.21	0.01	20
	Lowest methylation in promoter	*TRIM71* (Tripartite Motif Containing 71, LIN41)	0.15	0.02	3
		*CAPS2* (Calcyphosine 2).	0.26	0.005	12
		*UBE2QL1* (Ubiquitin Conjugating Enzyme E2 Q Family Like 1)	0.3	0.007	5
		*GLIPR1L2* (GLIPR1 Like 2)	0.3	0.008	12
PDL1 high/negative	Difference * in gene methylation	*S100A7L2* gene, also known as S100 Calcium Binding Protein A7 Like 2	0.74/0.57 Delta 17%	0.002	1
	Difference * in promoter methylation	*SOD1P3* (Superoxide Dismutase 1 Pseudogene 3),	0.68/0.54 Delta 14%	0.02	3
PDL1 negative/high	Difference * in gene methylation	*LINC00528* (Long Intergenic Non-Protein Coding RNA 528)	0.73/0.54 Delta 19%	0.001	22
	Difference * in promoter methylation	*NUMB* gene, (NUMB Endocytic Adaptor Protein)	0.51/0.31 Delta 20%	0.0005	14
PD-L1 Expression	Methylation status	Gene	Methylation	*p* value	Chr

* Delta beta values (mean.mean.diff).

## Data Availability

Data are available from the corresponding author on request.

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
