# Peer review of "Genome-Wide Methylation Analysis in Two Wild-Type Non-Small Cell Lung Cancer Subgroups with Negative and High PD-L1 Expression"

_cancers, 2024, doi:10.3390/cancers16101841_

Round 1

Reviewer 1 Report

Comments and Suggestions for Authors

The introduction outlines the significance of non-small cell lung cancer (NSCLC), emphasizing survival rates, precision medicine, PD-L1's role in immune therapy, and the rationale for conducting a methylation analysis. It suggests that the introduction requires better organization, transitioning smoothly from general lung cancer information to specifics about treatments and survival rates. The text recommends updating references to ensure scientific accuracy, using current nomenclature for genes, mutations, and therapies. It identifies a need for a more detailed discussion on how methylation impacts PD-L1 expression and the study's objectives.

1. A more explicit statement of the study's hypothesis and objectives at the end of the introduction is advised to better guide the methods and results sections.

2. It is recommended that the introduction be organized into well-defined sections and provide more scientific detail on methylation and PD-L1 expression.

The "Materials and Methods" section highlights justifications for sample selection and size, detailed protocols, and methodological choices as areas needing improvement. It suggests providing more information on the criteria for PD-L1 status determination and addressing potential inter-observer variability.

1. Clarification on the methods for increasing tumor content via microdissection and details on DNA extraction and mutation analysis are recommended.

2. The text lacks detail on next-generation sequencing (NGS) parameters and needs more information on data processing and quality control in computational data analysis.

3. Recommendations include enhancing methodological details, justifying choices, and addressing reproducibility concerns in the "Materials and Methods" section.

The "Results" section is discussed for its insights into differential methylation patterns in NSCLC. It notes a need for clarification on excluded methylation sites and statistical analysis. It calls for a deeper discussion of the biological implications of identified methylation patterns and the relevance of specific genes to PD-L1 expression.

1. Suggestions for the "Results" section include detailed statistical explanations, clarification of biological relevance, and improvement of interpretability.

The discussion is praised for integrating the study's findings within the broader context of cancer epigenetics but needs more detailed comparisons with existing literature. It is recommended that the study's limitations be examined more critically and that methylation's role in PD-L1 expression and tumor behavior be thoroughly explored.

1. To strengthen the section, direct integration of data from figures and tables into the discussion and outlining future research directions are suggested.

The conclusion summarizes the manuscript's contributions and suggests enhancing specificity, critical analysis, and articulation of future research directions in the discussion.

Comments on the Quality of English Language

non

Author Response

Subject: Appreciation for Your Review

Hello

I wanted to express my sincere gratitude for the time and effort you dedicated to reviewing my work. Your insightful comments and thoughtful feedback have been immensely valuable. Your suggestions for specific changes were spot-on. We have incorporated them into the revised manuscript. You will find the updated text in the attached file. Once again, thank you for your thorough review. Your expertise has significantly improved the quality of our work. If you have any additional insights or questions, please feel free to reach out.

Best regards,

Hutarew

Reviewer 2 Report

Comments and Suggestions for Authors

The authors analyzed 20 cases of lung adenocarcinomas and found that high PD-L1 expression is significantly correlated with an altered methylation pattern.

1.        How were cases selected? Were they consecutive?

2.        Lines 67-69. Were database searches used only to select genes to analyze or also to retrieve results? I think it is important to underline that your study is not based on internet databases.

3.        PD-L1 low is usually referred to low expression, i.e. 1-49%. Do the authors mean PD-L1 very low, almost negative <1%?

4.        In PD-L1 high expression, some genes were hypermethylated whereas other genes were hypomethylated. Could the authors explain better this difference? Were most of genes hypermethylated and only a minority of genes hypomethylated? Or do hypomethylation involve only secondary genes, so they were not important? The final message in the abstract, if I understand correctly, is that PD-L1 high expression shows hypermethylation in a considerable number of genes.

5.        A limitation of the study is a relatively low number of cases; this could be mentioned in the Discussion.

Author Response

(The authors gave the same response as above.)

Round 2

Reviewer 2 Report

Comments and Suggestions for Authors

The authors successfully addressed all points.

Congratulations for your work.